# mHealth: providing a mindfulness app for women with chronic pelvic pain in gynaecology outpatient clinics: qualitative data analysis of user experience and lessons learnt

Elizabeth Ball [1,2,3] Sian Newton,[2] Frank Rohricht,[4,5] Liz Steed,[6] Judy Birch,[7] Julie Dodds,[2] Clara Cantalapiedra Calvete,[1] Stephanie Taylor,[6] Carol Rivas [8]

► http://dx.doi.org/10.1136/bmjopen-2019-030164

For numbered affiliations see end of article.

**Correspondence to**
Dr Carol Rivas;
c.rivas@ucl.ac.uk

## ABSTRACT

**Objectives** To determine whether a pre-existing smartphone app to teach mindfulness meditation is acceptable to women with chronic pelvic pain (CPP) and can be integrated into clinical practice within the National Health Service (NHS) CPP pathways, and to inform the design of a potential randomised clinical trial.

**Design** A prestudy patient and public involvement (PPI) group to collect feedback on the acceptability of the existing app and study design was followed by a three-arm randomised feasibility trial. In addition, we undertook interviews and focus groups with patients and staff to explore app usability and acceptability. We also obtained participant comments on the research process, such as acceptability of the study questionnaires.

**Setting** Two gynaecology clinics within Barts Health NHS, London, UK.

**Participants** Patients with CPP lasting ≥6 months with access to smartphone or personal computer and understanding of basic English.

**Intervention** The intervention was mindfulness meditation content plus additional pain module delivered by a smartphone app. Active controls received muscle relaxation content from the same app. Passive (waiting list) controls received usual care.

**Main outcome measures** Themes on user feedback, app usability and integration, and reasons for using/not using the app.

**Results** The use of the app was low in both active groups. Patients in the prestudy PPI group, all volunteers, were enthusiastic about the app (convenience, content, portability, flexibility, ease of use). Women contributing to the interview or focus group data (n=14), from a 'real world' clinic (some not regular app users), were less positive, citing as barriers lack of opportunities/motivation to use the app and lack of familiarity and capabilities with technology. Staff (n=7) were concerned about the potential need for extra support for them and for the patients, and considered the app needed organisational backing and peer acceptance.

**Conclusion** The opinions of prestudy PPI volunteers meeting in their private time may not represent those of patients recruited at a routine clinic appointment. It may be

## Strengths and limitations of this study

► The study was designed with the help of a group of patients with chronic pelvic pain.
► Patient recruitment to the study was good.
► Our study focused on a deprived urban area in the UK and considered typical local clinical patients, which is not commonly done.
► The qualitative evaluation included the perspectives of both the patients and a variety of healthcare staff.
► Patients in the qualitative evaluation preferred telephone interviews over the offered face-to-face focus groups.

more successful to codesign/codevelop an app with typical users than to adapt existing apps for use in real-world clinical populations.

**Trial registration number** ISRCTN10925965.

## INTRODUCTION

Smartphone health apps, as one form of mobile health (mHealth),[1] are popular in the UK, which is our study setting. With more than two-thirds of the UK population using smartphones,[2–4] health apps are one of the fastest growing app categories, and the number of users is still increasing.[5] Currently these apps are usually developed either by researchers or (in the majority) by commercial companies, without collaboration between these groups.[6 7] The lack of interaction between researchers and commercial developers in the field of pain-related apps has led to a situation where commercially available apps have not been scientifically validated and apps that have been developed from research projects are not commercially available.[8]

We were interested in using an app to support women with chronic pelvic pain (CPP) in a clinical setting, where validation

of an intervention is important to ensure best care. CPP is defined as a subjective physical and emotional experience of pain in the pelvic area that has been present for at least 6 months and which may or may not have an identifiable pathology.[9] CPP affects up to 24% of women worldwide[10] and accounts for 20% of gynaecological clinic referrals.[11 12] It has considerable impact on patients' quality of life, including their mental health and income[13] due to loss of working days and diminished work capacity. The annual cost to the National Health Service (NHS) has been estimated at approximately £326 million.[14] For endometriosis alone, which is only one cause of CPP, a European study of over 900 women showed an average annual total cost per woman of €9579. The cost of productivity loss of €6298 was double the healthcare cost of €3113 per woman. The latter was due to surgery (29%), monitoring tests (19%), hospitalisation (18%) and physician visits (16%).[15]

Despite costly interventions, CPP is often resistant to surgical and medical treatment and appears to respond better to a multimodal, holistic approach,[16] with a focus on coping strategies. A systematic review of randomised controlled trials (RCTs) has identified mindfulness meditation (MM) as an effective coping strategy in other chronic pain conditions.[17] In addition, evidence from uncontrolled trials suggests positive effects of MM on CPP, such as an increased ability to control pain, and improvements in mental health, emotional well-being, work, and family life and social functioning.[18 19] These however have never been examined in an RCT.

We therefore chose to evaluate MM delivered via an app to women with CPP as our intervention. CPP is especially common in younger women, who may be categorised as digital natives, making an app-based intervention particularly appropriate to this group.

In MM the aim is to stay focused on one's own breathing. Whenever attention wanders to intrusive thoughts and feelings, these are simply taken notice of in a neutral way, that is, without elaboration or judgements or consideration of action. They are then let go as attention is returned to breathing. The idea is to bring awareness back to the here and now whenever worries and troubles intrude into thoughts.[20]

Generally two main complementary approaches have been used for MM: (1) exercises focusing one's attention to the present moment and (2) monitoring of experiences in the present moment. While systematic reviews show that MM may have positive effects on depression, quality of life and pain symptoms in patients with chronic pain,[17 21 22] and that apps with an MM focus have been shown to be beneficial for various chronic conditions,[23] none of the reviewed papers included meditation delivered via mobile phone apps or in women with CPP.

Evaluation of an existing app is considered to often be appropriate[24] and is both quicker and more cost-effective than designing an app from scratch. We chose to evaluate an existing commercial app platform that teaches mindfulness by guided meditation, with a 10-day

basic meditation module followed by a pain module specifically designed for the MEMPHIS (Mindfulness meditation using a smartphone application for women with chronic pelvic pain) study. The Headspace app we used was publicly nominated the favourite health app of 2013,[25] has a five-star user rating in the Apple app store, and has scored top in a systematic review of 23 mindfulness apps using the Mobile Application Rating Scale (visual aesthetics, engagement, functionality or information quality).[26] Headspace had reportedly seen over 15 million downloads up to mid-2018, when our study began.[27] To our knowledge the Headspace app in its original or modified form had not been assessed in any other pain conditions.

We undertook a three-arm, parallel, randomised feasibility trial (MEMPHIS)[28] to assess whether or not to proceed with a full RCT of the modified Headspace meditation app for women with CPP. In the current paper we report on the qualitative interview and focus group data from this study; the protocol and quantitative results have been published/submitted.[29 30] Our aim for the qualitative part of this study was to determine whether a pre-existing smartphone app to teach MM is acceptable to women with CPP and can be integrated into clinical practice within the NHS CPP pathways. The objectives were to consider:

► Acceptability, use and usability of the app in the intended service user population and for healthcare professionals (doctors, healthcare assistants, clinic and research nurses).
► Feasibility of integrating such an app into existing healthcare pathways.
► Usefulness of having a distinct patient group to advise us on the study design.

## METHODS
### Outcomes
The outcomes of this analysis were inductively derived descriptive themes on acceptability, use and usability of the app, and feasibility of integrating the app into existing pathways. We follow the ISO 9241-11 (International Organization for Standardization; https://www.iso.org/obp/ui/#iso:std:iso:9241:-11:ed-2:v1:en) concept of technology usability (user-friendliness) as the extent to which the app could be satisfactorily used by participants to meditate. By acceptability we mean whether participants could see a reason for using the app when given in the clinic and would be happy to use it for meditation.

### Intervention procedures
Women in the MM group received access to a 60-day progressive MM course delivered via the Headspace app. The first 10 days of the course taught the basics of MM. Following this, participants were able to access the module on meditation which was targeted for chronic pain. This module had been specifically made for this study. The length of the session was 10 min for the first 10 days, 15 min up to day 20, and 20 min up to day 60.

The active control group received access to a series of muscle relaxation sessions. These sessions were identical every day, except that their duration increased to mirror the increasing duration of the meditation content being listened to by the intervention group. Usage data are reported elsewhere.[30]

## Patient and public involvement

We held a patient and public involvement (PPI) group workshop before the study to discuss the acceptability of the Headspace app and to help us design our study. Women attending the Royal London Hospital CPP clinic were invited to volunteer for a week of using the unmodified (normal commercially available) Headspace app (which did not have the pain module at the time we undertook our prestudy workshop) and then feed back on their experiences with the app in an evening discussion group. Women were not involved in the design of the modified app. The focus of the PPI group was on the use of the generic MM app. However, two patient representatives provided support from the study design stage through to recruitment, and to the interpretation of the results, and regularly attended trial management group meetings.

## Study recruitment and eligibility

The trial recruited patients from two outpatient gynaecology clinics within the Barts Health NHS Trust in two separate deprived areas of inner East London. Female patients with new or follow-up gynaecology appointments were assessed for eligibility by a researcher in the clinic, having been posted a patient information sheet. Women were eligible if they had been suffering from CPP for 6 months or more and had at least a basic understanding of the English language, sufficient to follow instructions, as assessed during discussion about the study for informed consent; in the event, no women were excluded on this basis. Women were excluded if they did not meet these criteria or they did not have access to a smartphone or personal computer, or were currently using the Headspace app (there were very few of the latter, according to the impression of the recruiting nurses). All patients gave full and informed consent to be randomised. All healthcare professionals and research nurses involved in the two clinics (the only eligibility criterion applied for staff) were also invited to take part in the feasibility study. Full enrolment data are provided in Forbes *et al.*[30] A key difference from women in the PPI group was that the focus of the feasibility study patients was on managing their pain, with the app given explicitly as part of their clinic management support.

For the analysis of quantitative data, 90 patients were allocated randomly in a 1:1:1 ratio to the MM app, a muscle relaxation app active control or the usual care arm (for full details, see Ball *et al.*).[29] Patients in the two active arms were asked to download the modified app in the clinic with support from a research staff member and were sent a questionnaire about app usability, an analysis of which is reported in a companion paper.[30] We used data from the app usability questionnaire to inform topic guides for the qualitative part of the study. This uncovered and outlined key usability issues to guide our semistructured interviews and focus groups with patients and staff.

All women in the intervention and active control arms were eligible for the qualitative component of the study, as well as all staff participating in the study.

## Within-study interviews and focus groups

All patients in the intervention and active control arms were invited to one of the two focus groups at their own study site after the 6-month follow-up. We offered telephone interviews as an alternative. Patients were asked to 'walk through' the app with researchers, articulating their thought processes while they did so and commenting on its different specific features.[31] Walkthroughs are often used in developing technologies such as mHealth. This helped to identify app usability issues or barriers to use of the app from users' points of view without the need for technical discussions. The results of the walkthrough, with comments on the different features specific to the usability of the intervention app used in our study, are shown in online supplementary appendix 1; walkthroughs were undertaken by two patients. Patients also discussed with us their experiences around app usability and acceptability.

Staff were invited to attend a staff focus group overseen by the patient representative and facilitated by a researcher. Consultants, healthcare assistants, clinic and research nurses, and representatives from the Pelvic Pain Support Network were eligible. In addition to considering app usability and acceptability, members of the staff focus group were asked about the ease of integration into existing NHS pathways. Part of the staff discussion was freeflowing with open-ended questions, which gave us patient-focused information on app acceptability, and part was structured using questions developed from the Normalisation Process Theory (NPT) toolkit as recommended by its developers.[32] For example, we asked whether staff could see a purpose for the app in clinical practice, as adding something different, which corresponds to the NPT toolkit question 'Participants distinguish the intervention from current ways of working'. Since we used a semistructured approach, questions were not rigidly worded. This helped us to consider the feasibility of integration of the app into practice. NPT is a theory of implementation practices that was initially developed for consideration of technology implementation and is commonly used.[32]

All data were audio-recorded at point of collection and were transcribed, with personal identifying data removed from the transcripts. Raw data were stored in a Primary Care Clinical Trial Unit database adhering to clinical trial standards.

## Analysis

Analysis of within-study focus groups and interviews was carried out blinded as to which study app was used

and deployed the immersion-crystallisation method.[33] Thus, the lead qualitative researcher immersed herself in the data, reading transcripts carefully, then writing down articulated or crystallised patterns or themes that related to the study's aims and research questions. These were discussed with another researcher from the team, and themes were modified as appropriate. This process was repeated until all the data had been examined and all patterns that had been observed were articulated, discussed and substantiated with exemplar extracts. This approach was considered appropriate since we had a small data set and we were not aiming to develop conceptual themes but rather to inform the design and development of an RCT for the modified app.

We used the Standards for Reporting Qualitative Research checklist when writing our report.[34]

## RESULTS

We screened 488 women between May and September 2016 for eligibility to participate in the study. After exclusions, 90 women gave full consent to participate and were randomised to the intervention arm (31 women), active control arm (30 women) or usual care arm (29 women).

### Demographics

Women in the main feasibility trial[30] had a mean age of 35 years, 66% were employed, and overall approximately 50% had stayed in full-time education until at least the age of 20, although the proportion was lowest in the intervention arm at 36.7%. Overall 44% were of white ethnicity, although the proportion was lowest in the intervention arm (35.7%) and highest in the usual care arm (53.6%). The second most common self-reported ethnic groups were 'Southern Asian' and 'Black'. Women in the intervention arm were most likely to have experienced CPP for 3–5 years (40.3% of this group) and women in the usual care group for over 10 years (42.9% of this group). More women had pain for longer than 2 years in the intervention arm than in either of the other two arms. All women reported high pain intensity, with a mean of 6.8–6.9 in the previous week (on a scale from 1 to 10).[30] These and other demographic data are reported in more detail in Forbes *et al*.[30] Our qualitative sample was taken from the two active arms and comprised 16% of trial participants and 23% of those eligible for the qualitative study. We did not record separate demographic data for the women in this smaller sample.

### Prestudy PPI group

The 10 women in the prestudy PPI group were self-selected local women who were familiar with using apps and focused on app use per se. They anticipated no technical issues for women in the trial, even those who were not used to apps. They considered that the Headspace app would be successfully adopted by patients taking part in the study, given that a smartphone, like CPP, is 'always with you'. They praised the flexibility of the app,

welcomed its portability and were unanimous in saying it was easy to slip off for 10 min when at work to use it. As a result, they found they could use it at times when they most needed pain relief as well as to prevent pain, and found the app helpful in relieving pain and stress. The group reported being able to meditate without the app, once they had tried it with the app; however, they still preferred to use the app because they found the voice soothing.

### App usage in the study

Patient usage of the app was less than expected from our prestudy PPI group discussions. Only 36% of meditation app patients and 46% of the active control patients used the app at least once.[30]

### Thematic analysis of within-study data

Qualitative data were obtained from 14 study patients; 12 preferred a telephone interview and 2 attended face-to-face interviews, with 1 participant at the university attached to one of the recruiting clinics and the other participant at the other recruiting hospital. Patients chose not to attend focus groups. Four of the patients were from the active control arm and 10 from the intervention arm. The two women we met face to face had both used the intervention and neither had progressed beyond the training stage, something that we cannot discount for other participants and which may help to explain reports of lack of effect on pain. Seven people attended the staff focus group: two recruiting nurses, three clinic nurses, one consultant and a representative from the Pelvic Pain Support Network.

The qualitative analysis revealed three main themes and four subthemes with regard to usability from all within-study interviews and focus groups combined:
▶ Familiarity and capabilities with app technology.
▶ Motivations to use the app.
  – Perceived benefits.
  – Relation to other therapies.
▶ Opportunities to use the app.
  – Technology issues getting in the way.
  – Life getting in the way.

These are explored in the following sections. As the PPI group data were not research data, we did not analyse them for themes.

While we initially combined active control and intervention groups in our analysis, we then looked for instances where there was a difference between these two groups. Only where we found this difference in any theme or statement have we specified which group women belonged to.

### Familiarity and capabilities with app technology

Around half of the patients were sufficiently familiar with technology and apps to be comfortable using the study apps. However, six participants (all using the intervention, which was more complex than the active control) reported difficulty because they were "not very good at

technology" (Patient 1002, intervention), or were unsure how to get started or use the app effectively without help.

> I am not good with technical some things that is why the problems I had arisen, right okay. So I consulted with my daughter and she helped me work it out… so I don't try everything. (Patient 1002, intervention)

One further patient (Patient 1001, intervention) was not used to technical app language; 'help' suggested emotional support to her, for example. Two more (one intervention, one active control) changed handsets and therefore did not continue with the app. In all cases these technical difficulties appeared to lead to abandoning of the app or to restricted use of its functionality.

Five patients having technical problems suggested possible solutions such as a 'class' or group for first-time users, a YouTube orientation video or a pictorial leaflet. This might include an introduction to meditation and mindfulness as well as the app itself. One woman commented: "If your market is targeting people who are not using apps then you are going to have to get together and find ways to do this" (Patient 1041, intervention); she also suggested we could get ideas from other apps on the market in this regard.

Given their experiences in the study, staff were concerned about the additional staff time needed to support women in using the app. This would sit in tension with one of the original rationales behind choosing an app as the mode of delivery, which was to increase the effective use of staff contact time with patients. Language barriers might compound problems.

### Motivations to use the app

Staff, although unaware of the low sustained app use in the study, felt it would be common sense to hold occasional motivating meetings with patients if the intention was for them to use the app long term. The patient data suggested the main motivators or lack of motivation for using the app in our sample, which could be drawn on in such meetings, and which we now consider.

### *Perceived benefits*

Three patients from the intervention arm said they only entered the trial to help others through research, as were already using alternative forms of pain control. They explained that this meant they were not motivated to actually use the app, perceiving the relative benefit to be small. The failure of such altruism to extend to using the app is a recognised phenomenon in clinical trials and has been called 'weak altruism'.[35] Thus, only one of these three patients persevered. Even though she was one of the women who experienced difficulties with the technology, she explained: "with something that is as soul destroying as the pain, it is important to help others off the back of other people's misfortune as it were" (Patient 1036, intervention). However, she wondered how relevant her data were.

> I took steps to improve my situation from a weight loss perspective as well and I've lost a lot of weight which has significantly helped not 100% but it is has significantly helped so I felt a bit fraudulent the last time filling in the forms because, so everything had improved so much so from the medical study perspective it was more about the weight loss than the app. So I felt a bit bad that I was still taking part. (Patient 1036)

There was no clear pattern regarding the impact of current pain on app use by patients. Six said they used it regardless of pain intensity—sometimes developing a daily routine—while four only used it when in severe pain or expecting to be (eg, during menstruation). This cyclical or intermittent use in some patients, which was irrespective of study arm, should be considered when looking at our main study outcomes.[30] The Headspace app requires regular use to learn and benefit from psychological techniques. To address this, healthcare professional alerts have been effective in other studies,[36] while Headspace only has a reminder function that the user can set. This feature was easily overlooked though it could be effective; as one patient said: "To be quite honest I used it [the app] a couple of times and then forgot. And then I [remembered the reminder function and] used it more frequently" (Patient 1036, intervention).

One patient said medication was not working but the app did, although she was not sure whether this was "mind over matter" (Patient 1065, intervention), which was her term for a placebo effect. Three others said it did not reduce their pain; all three were using the intervention app. The remaining patients considered other benefits were good reason for using the app even when they did not feel that it reduced pain intensity. Alternative or unanticipated benefits were not formally measured or taken into account in the study's effectiveness outcomes.[30] For example, 10 patients valued the way the app helped them to relax or destress or to focus and reassess their life; three of these patients specifically said they used it to induce a relaxed state to get to sleep. Notably the active control was a relaxation app; however, this benefit was also reported by many women in the intervention arm. One participant (active control) said she did not like the focus on pain per se as her condition impacted on various areas of her life. Even when the intervention app was positively received, women might stop using it because it was too powerful, and they had gained the change they wanted:

> I think it was day 3, I could see the change that was happening, I was able to speak up for myself…I can't explain it, even now I am getting emotional…it's just a lack of focus, I just needed direction. To try and put it into words. To me it meant so much that I have gone back to church…I use it outside of the app now I have got from it what was missing, so it's done something to me and for me which is very positive, and I may try it to lose weight but those positive vibes are still there. I can't go back to it because I did not

want to go any further because what I got at the time helped me to focus, to change my way of thinking. I used it for about two or three weeks. (Patient 1001, intervention)

Three patients from the intervention arm found the app put them more in tune with their bodies and their breathing (two of these were among those who also found the app destressing), while another found yoga better for that. Six patients, like the prestudy PPI group members, also learnt to use techniques from the app to alter their stress patterns without the app, having tried it for example in traffic or by sitting down and taking time out or for general relaxation. Again, this would impact on trial results.

### Relation to other therapies

Two patients (both active control) preferred 'pure' meditation, while another considered the app to be "very much about meditation" (Patient 1041, intervention), which is in keeping with the arms they were in. An alternative therapy practitioner and two further patients reported that they preferred yoga. One (active control) said this was because it focused on each part of the body in turn.

Three patients from the intervention arm thought the app was useful as an adjunct to other methods rather than a replacement, for example physical interventions such as Pilates or listening to classical music.

### Opportunities to use the app
#### Technology issues getting in the way

Staff pointed out that not all patients had smartphones (not appreciating that personal computers/tablets were alternatives allowed in this study). Some patients lacked the storage space to load the app on their phones. There were also issues with Wi-Fi connectivity when the staff tried to help the patients load the app within the hospital sites. Possible solutions that the staff suggested were to lend patients phones and to have group upload sessions in a location with a good Wi-Fi signal, although they acknowledged the resource implications.

### Life getting in the way

Seven patients revealed they preferred to use the app in the evenings due to other life commitments. This meant they did not always use it as a direct response to pain, reducing its potential for contemporaneous effect. One patient who used it in response to pain but only used it once or twice blamed this on having no spare time due to juggling work and children (active control). However, another patient (intervention) managed despite such commitments; the fact that she was in the intervention arm may have played a role.

### Barriers to integration for staff

Staff believed that the biggest barrier to clinical adoption of the app was a possible lack of support from the host organisation. It might also be hard to integrate the app within existing professional work practices if the staff in the position of offering the app to patients failed to see its relative advantage over other interventions. Collection of feedback on the app's effectiveness would be necessary for staff to support sustained use. It was felt that the staff would need training on how to introduce the app to women in practice, and that complexity and high staff turnover could impede sustained use. An app was also seen as impersonal compared with face-to-face contact, which was more favoured by the staff.

### Participant comments on the research process

The study questionnaires that were used for the main quantitative outcome measures[30] were acceptable to patients except for some discomfort with a question about sex, which the patients considered a delicate question that was missing a 'no sex' option. Most preferred a paper form, reflecting their lack of affinity with technology. There were no indications that the study design or the study processes had contributed to participants' lack of engagement with the apps, with a caveat around support with the technology as mentioned above, although we did not systematically consider this. A full summary of patient comments on the study design and procedures is given in online supplementary appendix 1.

### DISCUSSION

Our study adds to the limited evidence on mHealth app user behaviour and experience.[36 37] The prestudy PPI group (young women, of a generation familiar with using apps and who were asked to focus on the study design use of the app) liked the idea of delivering the intervention via an app, praising the contemporary design and flexibility. Hence we expected a similar positive attitude from trial participants, who were of a comparable age and we assumed would be keen on using apps. Participant feedback revealed that this assumption was too simplistic.

Using our qualitative data, we were able to explore the reasons for low app usage that had been recorded in our feasibility study.[30] Our thematic analysis suggests that the low app use in the trial was because many patients were not familiar with apps in general or lacked capabilities with technology. This was particularly true for the more complex intervention app. The other themes we report did not differ between groups (although the three cases of 'weak altruism' all occurred in the intervention arm), which suggests more generic issues that can be transferable to other app use situations. For example, women stated limited motivation to use the app because of a lack of perceived benefit, or a lack of opportunity to use the app due to Wi-Fi issues or due to other commitments.

Similar findings were reported by Laurie and Blandford,[38] who interviewed 16 healthy city-dwelling participants (25–38 years) about their behaviour before and after 30–40 days of Headspace app exposure. Similar to our study, they reported barriers such as busy lives, failure to establish a routine and a lack of perceived benefit; all users in their study tried the app at least once hoping it

could deliver a quick fix but were disappointed if this did not happen. In our study many patients failed to perceive a benefit from using the app. Hence excuses stating other commitments may mask a deeper lack of motivation linked to perceptions of benefits.[39]

The advantages and disadvantages of using the app as a stand-alone were also illustrated by our data. Some suggestions made by participants to improve usage, such as more guidance at the start, seem obvious in hindsight. But they had not been considered because of the feedback from the prestudy PPI group and the commercial success of Headspace. The use of community contacts may be a helpful alternative.[40] Social support can create a community of practice, help to clarify expectations[41] and improve health outcomes (as shown, for example, in internet-based psychological treatment for depression).[42]

The data suggest that for successful app use we need to understand what motivates individuals with clinical need to use the app for clinical reasons (which our PPI group did not focus on) and target this, for example by setting appropriate expectations. Incentivisation might also improve motivation. This could be achieved through app gamification,[7] or encouragement through integration with patient–clinician face-to-face encounters, which was lacking in our study since the app was used as a stand-alone. The present study provided extensive initial technical support but no coaching and incentivising, in keeping with the protocol. Future app studies should take this into account. Participants in our study may also have benefited from training and support to improve their app use capabilities and guidance on how to create more opportunities for app use, such as through sharing experiences in clinic support groups. This is in keeping with the COM-B model of behaviour change,[43] which matched our themes, although this was only realised after the analysis. The COM-B model states that capability, opportunity and motivation are key drivers of behaviours. This has been used to develop a number of complex interventions including smartphone apps.[3]

Lack of engagement after recruitment, or good initial engagement but minimal or inconsistent use, has been reported in other studies.[44–47] This includes two Headspace trials other than our own,[46 47] set in a university and a psychiatric inpatient clinic, and both in the USA. Inconsistent app use was noted by Wen *et al*[48] among junior doctors who used self-guided Headspace. Morrison Wylde *et al*[49] compared face-to-face MM with Headspace use in novice paediatric nurses. However, unlike our study, there were no recorded dropouts/non-users and also no record of whether or how long the app was used, which is an important omission.

None of these studies included a qualitative component. Yet each of the Headspace study groups was very different, and so will likely have differed in motivations, contexts for opportunity to use the app and incentivisations.[50] While these aspects were not considered in the other studies, our use of qualitative research has enabled us to explore these in more depth. Our findings suggest

these are important considerations in any study of app use, and therefore this study makes a contribution to the field. For example, some of the groups in other studies may have differed from ours in the likelihood of using mHealth apps in the first place and familiarity with technology. Inpatients may have more time to use the app and more support, and may also have had specific barriers to the use of apps, such as related to setting and to illness.

Patients in the qualitative part of our study tended not to use apps on a regular basis (or at least apps other than simple games), and in terms of our themes, also represented in the COM-B model, may be said to have few capabilities in technology use. They therefore do not represent the typical users of the Headspace app in a commercial setting. Accessing the app regularly requires energy, time and effort, but patients with CPP often suffer from fatigue and anxiety as comorbidities, perhaps while having to juggle family life and work. Therefore, this may be seen as a challenging clinical population in which to trial an app. Further Headspace trials in outpatients with diabetes (NCT03274362) and pain (NCT03495726) are under way.

Our study has also shown that clear consideration of unexpected benefits should be included in future studies, and these can be informed by our finding that benefits for patients may be more diffuse than anticipated (eg, app relieving stress rather than pain). Such benefits were found in the active control as well as the intervention arm, and so it may be that they represent a placebo effect, although the effect could equally be real. Our data also suggest that staff benefits may be less than anticipated, as participants sometimes needed a lot of support and scaffolding in technology use at least initially.

Young age, comorbid anxiety and low educational attainment are predictors for dropping out of web-based interventions, according to studies in the field of depression.[51 52] This may be true despite regular phone support,[52] although our participants all considered active motivational support from staff or app support groups would have improved app use. Our intervention arm participants were particularly likely to be young and with low educational attainment.

Our data suggest that it is important to involve real-world end users in the agile design or development or modification of apps, in close collaboration with researchers and commercial app developers.[7] Although the evaluation of existing apps has been recommended as a cost-effective and rapid process,[24] our findings suggest that in actual clinical practice these may be problematic.

### Strengths and weaknesses of the study and in relation to other studies

One strength of this study is that it provides much-needed evidence in the field of evaluating existing health apps in a clinical population[6 8 24] and recording user experience. This provides us with lessons to be learnt.

Researchers conducting the interviews and focus groups included a senior mixed methods medical sociology researcher, a recruiting nurse, a representative from the Pelvic Pain Support Network and an experienced health psychologist. Findings were similar across the data, therefore the different backgrounds of the researchers do not appear to have influenced the findings. The main analysis was undertaken by a medical sociologist and so concordance with the COM-B model is not due to background discipline bias.

We were able to successfully recruit participants, and we obtained valuable information from patients with CPP who were recruited from a deprived urban area in the UK and who were typical local clinical patients.

However, we report a marked discrepancy between the attitudes of the prestudy PPI group of volunteer patients from the local area, who actively put themselves forward for a 7-day trial of the app, and the participants asked to take part when they attended clinics. The opinions of prestudy PPI volunteers meeting in their private time may not be representative of the opinions of patients recruited at a routine clinic appointment. Women in the PPI group were used to using apps, which had led them to be interested in the study in the first place. Whereas women in the PPI group had all trialled the app at home and work without support from us, many patients from clinics were unable to use their phone beyond calls, texts and photos. Moreover, most of the women we interviewed used the intervention app. We can only speculate as to why this is so, but it does mean that concordances and divergences across the intervention and active control arm do need to be treated with circumspection.

To our knowledge the present observation on failure of the PPI work to translate into practice in a trial has not been formally reported before, and is lacking from a recent comprehensive systematic review.[53] PPI is a stipulated requirement when applying for some funding, and the present research findings should be taken into account when drafting guidelines for future PPI. PPI groups are able to provide significant help and advice in any study, but our findings show the value of adding agile codevelopment as a requirement for app intervention development as likely to provide a more effective intervention than one informed by PPI alone. Moreover, there is a difference between app use for active clinical management (as with our study participants) and consideration of the potential use of app for this (as with our PPI group).

### Implications for clinicians and policymakers

Given the patchy use of the app and the way that some participants did not manage to unlock its full functionality, and an indication of diffusion of benefit, more work is needed to see whether the app reduces pain per se. This study is a good example of the need to move away from 'one size fits all' behavioural interventions. Future studies should do more work on implementation before doing an effectiveness trial. This will enable researchers to be more nuanced about saying who the app is effective for, if at all.

Strategies to involve busy, less motivated and less technologically experienced individuals in PPI and lay app design groups need to be further developed. These groups should include considerable scaffolding, which we have shown extends to study involvement by patients. More care is also needed to obtain PPI input that is representative of the target group, taking into account their capabilities, opportunities and motivational aspects. It may be useful to give the PPI group a small condition management task that emulates what trial participants will be required to do. Moreover, we can confirm a recent review suggesting that health apps should be codesigned with users,[7] rather than presenting them with a pre-existing app. These implications in our study are also generalisable to other studies on technology.

**Author affiliations**
[1]Department of Obstetrics and Gynaecology, Barts Health NHS Trust, London, UK
[2]Centre for Women's Health, Institute of Population Health Sciences, Queen Mary University of London, London, UK
[3]Centre for Maternal and Child Health Research, City University London, London, UK
[4]Centre for Psychiatry, Wolfson Institute for Preventive Medicine, Queen Mary University of London, London, UK
[5]East London NHS Foundation Trust, London, UK
[6]Centre for Primary Care and Mental Health, Institute of Population Health Sciences, Queen Mary University of London, London, UK
[7]Pelvic Pain Support Network, Poole, UK
[8]UCL Social Research Institute, University College London, London, UK

**Acknowledgements** We would like to thank all the researchers, consultant obstetricians and gynaecologists, and data assistants at each of the recruiting clinics for their hard work in promoting the study, recruiting participants and for data entry. Our thanks go to the Trial Steering Committee, Andrew Horne, Sohinee Bhattacharya, Christina Liossi and Hulya Guzel, for their constant support throughout the trial. We thank the Pelvic Pain Support Network and Endometriosis UK for their promotion and guidance in developing the study design. We would also like to acknowledge the NIHR RfPB programme for their ongoing support. Lastly, thank you to Headspace for providing our participants with access to the Headspace platform, designing novel content for the study, and for their continuous support and advice throughout the study.

**Contributors** EB led the study as the CI. EB and CR were the main authors of the grant application for this study, and are co-lead authors of the current paper. FR, ST, JD, JB, SN and LS contributed to study design and initial protocol. All authors provided support throughout the trial and contributed to the final paper. CR led on the PPI, and CR and LS led the interview and focus group field work and analysis reported here. CR, LS, SN, CCC, JD and JB were all involved in the field work.

**Funding** The UK National Institute for Health Research, Research for Patient Benefit (RfPB no PB-PG1013-32025) funded the MEMPHIS study.

**Disclaimer** The funder had no role in the study design, in the collection, analysis and interpretation of the data, in the writing of this report, or in the decision to submit the paper for publication. The first and last authors vouch for the integrity, completeness and accuracy of the data and analyses, and for the fidelity of this report to the protocol and statistical analysis plan. The views and opinions expressed herein are those of the authors and do not necessarily reflect those of the RfPB, NIHR, NHS or the Department of Health.

**Competing interests** None declared.

**Patient consent for publication** Not required.

**Ethics approval** The MEMPHIS trial was a three-arm, parallel, randomised feasibility trial approved by Camden and Kings Cross Research Ethics Committee in 2016 (15/L0/1967).

**Provenance and peer review** Not commissioned; externally peer reviewed.

**Data availability statement** No data are available. The data are collected from a small number of people which could compromise their identity if shared with others. Therefore we are not making them available except under exceptional circumstances, which will be determined by the custodian of the data (EB) on an individual basis.

**ORCID iDs**
Elizabeth Ball http://orcid.org/0000-0001-8739-090X
Carol Rivas http://orcid.org/0000-0002-0316-8090

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
