## [Reviewer comments · BMJ Open]

ARTICLE DETAILS

TITLE (PROVISIONAL)	Mhealth – Providing a Mindfulness App for women with chronic pelvic pain in gynaecology outpatient clinics: Qualitative data analysis of user experience and lessons learnt
AUTHORS	Ball, ELizabeth; Newton, Sian; Rohricht, Frank; Steed, Liz; Birch, Judy; Dodds, Julie; Cantalapiedra Calvete, Clara; Taylor, Stephanie; Rivas, Carol

VERSION 1 – REVIEW

REVIEWER	Sarah Martin UCLA, USA
REVIEW RETURNED	24-May-2019

GENERAL COMMENTS	The manuscript “Mhealth – Using a Mindfulness App for women with chronic pelvic pain: Qualitative data analysis of user experience and lessons learnt,” describes the qualitative assessment of the usage of a mindfulness-based mobile app. a commercially available app the authors. The assessment of a commercially available app and collecting qualitative data to examine barriers and motivations for using an app-based intervention are definite strengths of this manuscript. However, one significant limitation is that intervention and control data are not separated which considerably limits the interpretation of the results. Please see my additional comments below. General Concerns/Comments: 1. The collapsing of interview data across app groups is a significant concern. If the objective of the study was to examine the acceptability of the modified Headspace app, I think it would be more appropriate to analyze themes from the Headspace app group separately. The inability to determine whether the responses are in reference to the intervention or the control makes the interpretation of results in a meaningful way very difficult.2. The overall organization of the manuscript needs attention. The manuscript would be greatly improved if details about the aims, measures/outcomes, procedures, participants, and intervention were clearly outlined in the methods section.a. For example:i. At the end of the Introduction, please consider stating each aim/objective of the current study and define what determines “whether or not to proceed” with a RCT. Statements throughout the introduction reference the objectives of the current
--

	study, but concisely stating the objectives and specific aims of the study at the end of the introduction would improve clarity. ii. Was the PPI group part of the aims of the study? If so, this needs more clarification and specific outcomes of this group need to be listed iii. The authors mention collecting data from both patients and staff. Eligibility criteria and recruitment procedures for both patients and staff should be included in the participants section of the methods. iv. Please include a clear definition of usability, acceptability, and feasibility and how these factors were assessed through the qualitative methods. 1. Similarly, the methods section could be improved by adding a measures/outcomes section that lists and describes the qualitative outcomes used for the analyses. v. Please consider adding a section that describes the intervention. The authors mention an unmodified and modified version of the app and it is unclear how these versions differ and how the modified app was created. 3. As this is a qualitative analysis, the authors should consider the value in including quantitative information in the results/discussion (i.e. reporting numbers of participants who expressed various responses). Introduction 4. Please consider revising the first sentence of the second paragraph to first state the significance of this study and then state the objective. 5. Provide a citation for the second sentence on page 5, lines 5-7. 6. Pg. 5, lines 12-14: Please elaborate on the relationship between CPP, income, and annual costs to the NHS. Is this because of work missed due to symptoms? Appointments? 7. Pg. 5, lines 18-31: I think this section could be strengthened and clarified a bit. Please consider rephrasing, and better describe the cited “positive effects” of mindful meditation (i.e. are these positive effects on pain? Functioning? Quality of life?). On page 5, line 22, it is unclear what “focus” is referring to. 8. Please provide a citation for the mindfulness definition (page 5, lines 34-38). 9. Please consider moving the description of the Headspace app to the methods section. Has Headspace been examined in other pain studies? Mention of those could be relevant. 10. Spell out the MEMPHIS acronym the first time it is used. Methods 11. See comments under “General Comments.” 12. Further clarification of the PPI procedures is needed. What was the eligibility criteria for these participants (both the women attending clinics and the patient representatives)? What type of feedback was collected? Please describe the Trial Management Group meetings. 13. How was “basic understanding of English” determined? Please consider revising this phrase. Did participants need to be fluent in English? 14. On page 7, lines 16-18, the phrase, “there were very few of the latter” is vague. Please provide specific enrollment percentages.
--	---

	15. A description of the three arms is needed. 16. Please describe the randomization procedures. 17. A brief description of the intervention procedures (e.g., how long patients used the app, how often they used the app, what was included in the app, etc) is needed. 18. The statement about interview guides on page 7, lines 32-39, is hard to follow and it is unclear what data were used to develop the guides. Please consider expanding the description of this procedure, specifying the type of data, and describing how these data informed interview guide development in the "Interview and focus groups" section. 19. Please consider removing descriptions of the qualitative outcomes and analyses from the "Interview and focus groups" section (See comment iv-1 in General Comments). 20. In addition, describing the patient interview development and procedures first and then describing the staff interview development and procedures would improve clarity. 21. Please provide more justification for using the NPT toolkit and describe how it was modified. Analysis 22. Please provide a citation for last sentence of the first paragraph. Results 23. The information in the first paragraph should be moved to the methods section. 24. Please consider revising the demographic percentage reporting and include percentages for all arms and groups. Consider possibly including a table. 25. The collection of pain data is not mentioned in the methods section. Please include a description of these assessments in the methods section. 26. When stating differences across groups, please indicate the means or percentages and state whether or not these differences were statistically significant (including stats data). 27. Please consider using either "usage" or "adherence". These terms refer to different concepts and without knowing how often patients were instructed to use the app, "adherence" may not be an appropriate term to describe the app usage results. Did the authors consider assessing frequency of use? 28. In the thematic analysis section, please include the percentage of patients who participated and specify which arm these patients were in. 29. What were the reasons for patients not wanting to attend the focus groups? 30. Why did the authors analyze themes across app groups? See comment 2 in General Comments. 31. Please move the discussion of qualitative themes and observations to the discussion section. 32. The comments on the research process is not mentioned in the aims or methods sections. These procedures and assessments need to be explained prior to the results section. In addition, please consider rephrasing the statement on page 15, lines 55-58, ("There were no indication that...") as the authors cannot be certain that the study design or procedures did not affect engagement.
--	---

	Discussion 33. If the qualitative themes were derived from participants who used the Headspace mindfulness and the muscle relaxation app, please consider referring to both app types when discussing the results. 34. Paragraph 2: I'm concerned about statements like "many patients failed to perceive a benefit from using the app" when the results are unable to determine which app participants are referring to in their responses. 35. Pg. 17, lines 51-60: How does qualitative analysis provide for special consideration to contextual factors? 36. Pg. 18, lines 38-41: what indications specifically implied that many with benefits were in the intervention arm? 37. In the introduction, the authors state that an app-based intervention may be more appropriate for a younger sample, but the results suggest barriers. I wonder if the authors would want to discuss this further in the discussion. 38. Please consider revising the last paragraph of the discussion, as the current study did not aim to assess effectiveness. Thank you for the opportunity to review this manuscript.
--	---

REVIEWER	Christina Bryant University of Melbourne
REVIEW RETURNED	06-Jun-2019

GENERAL COMMENTS	This manuscript addresses an important clinical issue, that is both difficult to treat and under-researched. The authors start from the reasonable premise that evaluating an existing app would be more efficient than designing one from scratch. Consequently, although PPI activities were undertaken to comment on the study design, potential end users did not inform the design of the app. Recruitment was relatively straightforward, but actual app use was extremely low. The authors are honest in their recognition that end users need to be involved in the design of future apps, and I strongly support their conclusion that co-development of an app is required. This could ensure that the app's material really meets the needs of this group, rather than being generic in nature. The authors also make useful comments on the limitations of PPI activities if the role that participants are given do not allow sufficient input into the product. One aspect of the study that the authors should comment on is the decision to analyse the qualitative data blind to the which arm the participant was allocated to. If the purpose of the analysis was to gain an in-depth understanding of patterns of usage, reasons for non-engagement/discontinuation etc. surely it would be relevant to know which arm of the trial the comments apply to ? Could the authors please explain why they analysed the data blinded, and did not unblind after initial analysis? Overall I found this to be an honest description and discussion of the difficulties encountered in real life clinical research with a sometimes difficult client group. Although the outcomes were disappointing, the authors point out important implications for both clinicians and app designers.
---

VERSION 1 – AUTHOR RESPONSE

Responses to peer review 1	
General Concerns/Comments	
1. The collapsing of interview data across app groups is a significant concern. If the objective of the study was to examine the acceptability of the modified Headspace app, I think it would be more appropriate to analyze themes from the Headspace app group separately. The inability to determine whether the responses are in reference to the intervention or the control makes the interpretation of results in a meaningful way very difficult.	We agree and have unblinded the analysis. Where there is an impact on themes we have made this clear, e.g. by stating the groups and/or by making comments in the text.
2. The overall organization of the manuscript needs attention. The manuscript would be greatly improved if details about the aims, measures/outcomes, procedures, participants, and intervention were clearly outlined in the methods section. i. For example: At the end of the Introduction, please consider stating each aim/objective of the current study and define what determines “whether or not to proceed” with a RCT. Statements throughout the introduction reference the objectives of the current study, but concisely stating the objectives and specific aims of the study at the end of the introduction would improve clarity.	We have added the aim of the current paper at the end of the introduction. We added subheadings in the methods section specifying the details (aim, procedures, participants etc). The stop-go determination for the RCT is not relevant to this analysis, it is covered in our companion paper (Forbes et al, BMJ open submitted). The present paper deals with the qualitative aspects of crescendo, the companion paper with the quantitative aspects. We have made this clear now and agree it was confusing before.

ii. Was the PPI group part of the aims of the study? If so, this needs more clarification and specific outcomes of this group need to be listed	A PPI group as such was part of the study governance rather than an aim of the study. The PPI group was used pre-study to inform the study design as is desired by the NIHR Research for Patient Benefit funding scheme (our funders).
iii. The authors mention collecting data from both patients and staff. Eligibility criteria and recruitment procedures for both patients and staff should be included in the participants section of the methods	Eligibility criteria are clearly stated in the recruitment section but we have renamed this recruitment and eligibility for clarity and have clarified staff eligibility which we agree was less clear.
iv. Please include a clear definition of usability, acceptability, and feasibility and how these factors were assessed through the qualitative methods	We have defined usability, acceptability but not feasibility, because we are using it in the common everyday sense of the word. We have spelled out how these were determined in our interviews and focus groups section.
Similarly, the methods section could be improved by adding a measures/outcomes section that lists and describes the qualitative outcomes used for the analyses	In an inductive qualitative study the outcomes are the themes, but more than that cannot be specified. The objectives make our focus clear. Therefore we have not added text in response to this.
v. Please consider adding a section that describes the intervention The authors mention an unmodified and modified version of the app and it is unclear how these versions differ and how the modified app was created	We had previously said: ‘The intervention was mindfulness meditation content plus additional pain module delivered by smartphone app, active controls received muscle relaxation content by the same app.’ To clarify the second point we have modified the text to state: unmodified (normal commercially available) Headspace app for a week (which did not have the pain module at the time we undertook our study)
3. As this is a qualitative analysis, the authors should consider the value in including quantitative	This is our style preference. We, along with many other qualitative researchers, feel it gives information on the significance of a theme in terms of its commonality across participants. We do not

information in the results/discussion (i.e reporting numbers of participants who expressed various responses)	use it to indicate anything statistical and have not claimed this. Since this is common practice we prefer to keep it.
Introduction	
4. Please consider revising the first sentence of the second paragraph to first state the significance of this study and then state the objective	This paragraph is background information, we have added the objective at the end of the introduction
5. Provide a citation for the second sentence on page 5, lines 5-7	A reference has been inserted
6. Pg 5, lines 12-14: Please elaborate on the relationship between CPP, income, and annual costs to the NHS. Is this because of work missed due to symptoms? Appointments?	Explanation has been given for the relationship between CPP, income, and annual costs to the NHS
7. Pg 5, lines 18-31: I think this section could be strengthened and clarified a bit. Please consider rephrasing, and better describe the cited "positive effects" of mindful meditation (i.e are these positive effects on pain? Functioning? Quality of life?) On page 5, line 22, it is unclear what "focus" is referring to	This has been done Positive effects of mindfulness have been stated We agree and have amended the sentence, which now reads: 1) exercises focusing one's attention to the present moment and 2) monitoring of experiences in the present moment.
8. Please provide a citation for the mindfulness definition (page 5, lines 34-38)	We have changed the section and provided a reference.
9. Please consider moving the description of the Headspace app to the methods section Has Headspace been examined in other pain studies? Mention of those could be relevant	As the description is about popularity rather than processes we prefer to leave it where it is in the background as it helps to justify our decision to use it We have carried out a pubmed search on 4.8.19 and we have added the sentence on page 6 to the second paragraph stating "To our knowledge the

	headspace app in its original or modified form has not been assessed in any other pain conditions.”
10. Spell out the MEMPHIS acronym the first time it is used	This has been done- Page 6 line 9
11. Methods. See comments under “General Comments.”	Dealt with in previous responses
12. Further clarification of the PPI procedures is needed	We think this may already covered in our response to 2 ii. We are not sure what else would be required beyond our description in Patient and Public Involvement . We are happy to amend this if there is further clarification.
What was the eligibility criteria for these participants (both the women attending clinics and the patient representatives)? What type of feedback was collected?	These are specified in the paper; we have amended the text to make the differences more clear. Clinic attendance - The pre-study PPI group was recruited from the Royal London Hospital CPP clinic and the in-study patients (focus groups and interviews) were from Whipps Hospital and the Royal London Hospital CPP clinics. Evening discussion group on experience with the app and feedback on study process.
Please describe the Trial Management Group meetings	The TMG meetings are relevant to the companion paper only, as this is a qualitative report of post-trial work
13. How was “basic understanding of English” determined? Please consider revising this phrase	Added: sufficient to follow instructions as assessed during discussion about the study for informed consent; no women were excluded on this basis).
Did participants need to be fluent in English?	No
14. On page 7, lines 16-18, the phrase, “there were very few of the latter” is vague	We agree, but unfortunately recruiters did not record exactly how many – we have added ‘according to the impression of the recruiting nurses’
Please provide specific enrolment percentages	These are reported in our companion paper and we now reference this. We decided in an earlier draft to omit this to avoid repetition across papers
15. A description of the three arms is needed	We believe this is already in the paper, we state For the study of quantitative data, 90 patients were

	allocated randomly in a 1:1:1 ratio to the mindfulness meditation app, a muscle relaxation app active control or the usual care arm. And earlier we also describe these in more detail
16. Please describe the randomization procedures	We believe this is unnecessary detail for this qualitative paper and it is reported in our companion paper and our protocol which we now reference in this sentence
17. A brief description of the intervention procedures (e.g., how long patients used the app, how often the used the app, what was included in the app, etc) is needed	We have included the following paragraph on p7: Both apps ran over 10 minutes. The intervention app included an introductory module on the basics of MM, after which the participants accessed the pain module. The control app contained a ten-minute instruction of progressive muscle relaxation. Usage data is reported elsewhere (Forbes et al, BMJ Open submitted).
18. The statement about interview guides on page 7, lines 32-39, is hard to follow and it is unclear what data were used to develop the guides	We have amended this: We used data from the app usability questionnaire to inform topic guides for the qualitative part of the study. This outlined key usability issues that had been uncovered, to guide our semi-structured interviews and focus groups with patients and staff.
Please consider expanding the description of this procedure, specifying the type of data, and describing how these data informed interview guide development in the "Interview and focus groups" section	Again this was in an earlier draft but involved repetition of our companion paper and actually detracted from the flow and arguments in this paper, so having seen that it did not work, we prefer not to add it back. However we hope our slight change to the text is sufficient.
19. Please consider removing descriptions of the qualitative outcomes and analyses from the "Interview and focus groups" section (See comment iv-1 in General Comments)	Amended to The main focus was on app usability and acceptability.
20. In addition, describing the patient interview development and procedures first and then describing the staff interview development and procedures would improve clarity	We agree and have done so thank you.
21. Please provide more justification for using the NPT toolkit and describe how it was modified	We have amended as follows: Staff were invited to attend a staff focus group overseen by the patient representative and facilitated

	by a researcher. In addition to considering app usability and acceptability, members of the staff focus group (doctors, health care assistants, clinical and research nurses) were asked about the ease of integration into existing NHS pathways. Part of the staff discussion was free flowing with open-ended questions, which gave us patient-generated information on app acceptability, and part was structured using questions developed from the Normalisation Process Theory (NPT) toolkit in the way recommended by the NPT developers (26). For example we asked whether staff could see a purpose to the app in clinical practice, as adding something different, which corresponds to the NPT toolkit question 'Participants distinguish the intervention from current ways of working'. Since this was a semi-structured approach questions were not rigidly worded. This helped us to consider the feasibility of integration of the app into practice. NPT is a theory of implementation practices that was initially developed for consideration of technology implementation and is in common use (26).
Analysis	
22. Please provide a citation for last sentence of the first paragraph	We do not feel this needs a citation, it was a decision made by our experienced team members
Results	
23. The information in the first paragraph should be moved to the methods section	We prefer not to move this for the following reasons: Methods sections should not include any data collection figures, though this is sometimes done. These data are quite clearly results.
24. Please consider revising the demographic percentage reporting and include percentages for all arms and groups. Consider possibly including a table	We decided not to include the full data because they apply to the main study participants whose data are reported in full in our companion paper
25. The collection of pain data is not mentioned in the methods section Please include a description of these assessments in the methods section	That is because we have not detailed the feasibility study methods – but we felt the main demographic and pain data were important context in relation to our study. We have now cross-referenced to our companion paper and clarified these data are from the main study to avoid confusion.
26. When stating differences across groups, please indicate the means or percentages and state whether or not these differences	These data are provided in our companion paper and this is a qualitative paper

were statistically significant (including stats data)	
27. Please consider using either “usage” or “adherence” These terms refer to different concepts and without knowing how often patients were instructed to use the app, “adherence” may not be an appropriate term to describe the app usage results	‘Adherence’ has been replaced with ‘usage’ throughout the text
28. Did the authors consider assessing frequency of use? 28	Yes, this is reported in our companion paper. Reference to frequency assessment has been made on p7 in the added paragraph
29. In the thematic analysis section, please include the percentage of patients who participated and specify which arm these patients were in	Amended
What were the reasons for patients not wanting to attend the focus groups?	People said they had not the time or were not interested or did not use the app. As we did not systematically assess this we have not included it in the text.
30. Why did the authors analyze themes across app groups? See comment 2 in General Comments	Sorted, see above
31. Please move the discussion of qualitative themes and observations to the discussion section	We are unclear on this. Our non-theoretical description is of findings. We can see nothing in there that belongs in the discussion. We would need clarification.
32. The comments on the research process is not mentioned in the aims or methods sections. These procedures and assessments need to be explained prior to the results section	This has now been specified in methods
In addition, please consider rephrasing the statement on page 15, lines 55-58, (“There were no indication that...”) as the authors cannot be certain that the study design or procedures did not affect engagement	We have added: though we did not systematically consider this
Discussion	
33. If the qualitative themes were derived from participants who used the Headspace mindfulness and the muscle relaxation app, please consider referring to both	amended

app types when discussing the results	
34. Paragraph 2: I'm concerned about statements like "many patients failed to perceive a benefit from using the app" when the results are unable to determine which app participants are referring to in their responses	We have addressed this concern by unblinding
35. Pg 17, lines 51-60: How does qualitative analysis provide for special consideration to contextual factors? 36	We have now explicated this: While these aspects were not considered in the other studies, our use of qualitative research has enabled us to explore these in more depth.
36. Pg 18, lines 38-41: what indications specifically implied that many with benefits were in the intervention arm?	Amended as now unblinded
37. In the introduction, the authors state that an app-based intervention may be more appropriate for a younger sample, but the results suggest barriers. I wonder if the authors would want to discuss this further in the discussion	The first paragraph of the discussion has been amended
38. Please consider revising the last paragraph of the discussion, as the current study did not aim to assess effectiveness.	The paper now ends on a call for co design, which we believe is a key message

Responses to peer review 2	
One aspect of the study that the authors should comment on is the decision to analyse the qualitative data blind to the which arm the participant was allocated to. If the purpose of the analysis was to gain an in-depth understanding of patterns of usage, reasons for non-engagement/discontinuation etc. surely it would be relevant to know which arm of the trial the comments apply to? Could the authors please explain why they analysed the data blinded, and did not unblind after initial analysis?	We have unblinded the allocation now and reported results accordingly

VERSION 2 – REVIEW

REVIEWER	Sarah Martin UCLA
REVIEW RETURNED	11-Oct-2019

GENERAL COMMENTS	The resubmission of this manuscript improved significantly from the initial submission and it is evident that the authors took great care in addressing the reviewers' comments. The authors have adequately addressed most of my initial comments. My remaining comments are listed below: Comment in response to original comment #2: I appreciate the authors' edits and response. One remaining suggestion would be to move the list of objectives from the methods section to the aims section of the introduction. The specific objectives of the study should be provided prior to the methods section. Further, I think that a mention of the PPI group when presenting the study objectives would be beneficial as the recruitment and procedures of this group are reported on in the methods. I also feel as though the inclusion of this group is a strength of the study and it may be beneficial to readers to have an introduction to this procedure. Thank you for the opportunity to review this manuscript.
--

VERSION 2 – AUTHOR RESPONSE

Many thanks for accepting our revision.

We have made all the changes requested as indicated in the marked up copy.

There was one comment we did not understand, suggesting more detail on the PPI group: "I also feel as though the inclusion of this group is a strength of the study and it may be beneficial to readers to have an introduction to this procedure." We do in fact already describe this as well as we feel able to do. Does the reviewer mean us to reference PPI work more generally as a process in the introduction? We have however included PPI work as an objective, as suggested.

Unfortunately last week we accidentally submitted an old version rather than the version now submitted, and so it was rejected due to formatting errors. We have ensured that all formatting errors are corrected. Please let us know if anything else is required,